

# Patients with coronary heart disease, dilated cardiomyopathy and idiopathic ventricular tachycardia share overlapping patterns of pathogenic variation in cardiac risk genes

Christian Guelly[1,*], Zhannur Abilova[2], Omirbek Nuralinov[3], Katrin Panzitt[1], Ainur Akhmetova[2], Saule Rakhimova[2], Ulan Kozhamkulov[2], Ulykbek Kairov[4], Askhat Molkenov[4], Ainur Ashenova[4], Slave Trajanoski[1], Gulzhaina Abildinova (Rashbayeva)[3], Galina Kaussova[5], Christian Windpassinger[6], Joseph H. Lee[7], Zhaxybay Zhumadilov[2], Makhabbat Bekbossynova[3] and Ainur Akilzhanova[2,*]

[1] Center of Medical Research, Medical University of Graz, Graz, Austria
[2] Laboratory of Genomic and Personalized Medicine, Center for Life Science, National Laboratory Astana, Nazarbayev University, Nur-Sultan, Kazakhstan
[3] National Research Cardiac Surgery Center, Nur-Sultan, Kazakhstan
[4] Laboratory of Bioinformatics and Systems Biology, Center for Life Sciences, National Laboratory Astana, Nazarbayev University, Nur-Sultan, Kazakhstan
[5] Kazakhstan medical university "KSPH", Almaty, Kazakhstan
[6] Institute of Human Genetics, Medical University of Graz, Graz, Austria
[7] Sergievsky Center Taub Institute, Columbia University Medical Center, New York, NY, United States of America
[*] These authors contributed equally to this work.

Corresponding author
Ainur Akilzhanova,
akilzhan_ainur@mail.ru

## ABSTRACT

**Background**. Ventricular tachycardia (VT) is a major cause of sudden cardiac death (SCD). Clinical investigations can sometimes fail to identify the underlying cause of VT and the event is classified as idiopathic (iVT). VT contributes significantly to the morbidity and mortality in patients with coronary artery disease (CAD) and dilated cardiomyopathy (DCM). Since mutations in arrhythmia-associated genes frequently determine arrhythmia susceptibility screening for disease-predisposing variants could improve VT diagnostics and prevent SCD in patients.

**Methods**. Ninety-two patients diagnosed with coronary heart disease (CHD), DCM, or iVT were included in our study. We evaluated genetic profiles and variants in known cardiac risk genes by targeted next generation sequencing (NGS) using a newly designed custom panel of 96 genes. We hypothesized that shared morphological and phenotypical features among these subgroups may have an overlapping molecular base. To our knowledge, this was the first study of the deep sequencing of 96 targeted cardiac genes in Kazakhstan. The clinical significance of the sequence variants was interpreted according to the guidelines developed by the American College of Medical Genetics and Genomics (ACMG) and the Association for Molecular Pathology (AMP) in 2015. The ClinVar and Varsome databases were used to determine the variant classifications.

**Results**. Targeted sequencing and stepwise filtering of the annotated variants identified a total of 307 unique variants in 74 genes, totally 456 variants in the overall study
group. We found 168 mutations listed in the Human Genome Mutation Database (HGMD) and another 256 rare/unique variants with elevated pathogenic potential. There was a predominance of high- to intermediate pathogenicity variants in *LAMA2*, *MYBPC3*, *MYH6*, *KCNQ1*, *GAA,* and *DSG2* in CHD VT patients. Similar frequencies were observed in DCM VT, and iVT patients, pointing to a common molecular disease association. *TTN, GAA, LAMA2,* and *MYBPC3* contained the most variants in the three subgroups which confirm the impact of these genes in the complex pathogenesis of cardiomyopathies and VT. The classification of 307 variants according to ACMG guidelines showed that nine (2.9%) variants could be classified as pathogenic, nine (2.9%) were likely pathogenic, 98 (31.9%) were of uncertain significance, 73 (23.8%) were likely benign, and 118 (38.4%) were benign. CHD VT patients carry rare genetic variants with increased pathogenic potential at a comparable frequency to DCM VT and iVT patients in genes related to sarcomere function, nuclear function, ion flux, and metabolism.

**Conclusions**. In this study we showed that in patients with VT secondary to coronary artery disease, DCM, or idiopathic etiology multiple rare mutations and clinically significant sequence variants in classic cardiac risk genes associated with cardiac channelopathies and cardiomyopathies were found in a similar pattern and at a comparable frequency.

# INTRODUCTION

A total of 17.5 million people died from cardiovascular diseases in 2012 and of these, 7.4 million deaths were due to coronary heart disease (CHD), making it the number one cause of cardiac deaths, according to the WHO (*World Health Organization, 2018*). More than 75% of the cardiovascular disease (CVD) deaths occurred in low- and middle-income countries, with high rates in Kazakhstan, despite numerous state-run health programs. The age-standardized mortality rate of cardiovascular disorders in Kazakhstan is amongst the highest in the world. According to the latest WHO data published in 2018, there were 47,651 coronary heart disease deaths in Kazakhstan, amounting to 34.56% of total deaths. The age-adjusted death rate is 306.02 per 100,000 population, which puts Kazakhstan as eighth in the world, and approximately 9.6-fold higher than Japan (2018: 31.55 per 100,000 of population ranks Japan #181 in the world) and 4.05-fold higher than Austria (a death rate of 75.74 per 100,000 population ranks Austria #143 in the world) (*World Health Organization, 2018*).

Kazakhstan is located in the middle of Central Asia on the ancient Great Silk roads. Its vast territory covers 2,724,900 $km^2$ and it is the world's 9th largest country. Kazakhstan has a population of more than 17 million people (2016), with 131 ethnicities, including Kazakh (63% of the population), Russian, Ukrainian, German, Uzbek, Tatar, and Uyghur as the
most predominant groups. Historically, the ethnic Kazakhs were nomadic and migration led to the admixing of western and eastern tribes. Kazakhstan is ethnically and culturally diverse, which is due, in part, to the forced migration of settlers and mass deportations of ethnic groups, starting in the 19th century until the first third of the 20th century (*Lee & Akilzhanova, 2015*).

Genetic studies in Kazakhstan are challenging because of the genetic heterogenicity introduced by many ethnicities. However, data from diverse, heterogeneous populations exposed to the same environmental conditions and similar lifestyles yield important information on natural genetic plasticity. Such data are fundamental to genetic epidemiology and are critical to dissect natural polymorphisms from pathogenic alterations (*Akilzhanova et al., 2014*). Genome-wide data and linkage disequilibrium patterns are unavailable for Central Asian populations and are not represented in publicly available databases.

Cardiovascular disease encompasses a range of conditions from diseases of the vasculature, myocardial infarction, and congenital heart disease, most of which are heritable. Enormous effort has been invested in understanding the genes and specific DNA sequence variants responsible for this heritability (*Abraham et al., 2014*; *Al-Hassnan et al., 2016*).

Dilated cardiomyopathy (DCM) accounts for 30–40% of all heart failure cases and is a leading cause of heart transplantation (*Haas et al., 2015*). An autosomal dominant inheritance pattern of transmission and some autosomal recessive or X-linked recessive familial cases have been reported as indicated by the familial aggregation of DCM (30–50% of all DCM cases) (*Al-Hassnan et al., 2016*; *Haas et al., 2015*). Mutations in more than 30 disease genes have been linked as causative mechanisms of DCM (*Haas et al., 2015*; *Meder et al., 2011*).

In contrast, CHD, which is typically a result of coronary artery disease (CAD), is a complex disease driven by interactions between genetic factors and environmental stimuli and stressors. The genetic basis of CAD/CHD has been addressed by multiple genome-wide association studies (GWAS) enrolling thousands of individuals (*Lieb & Vasan, 2013*; *Consortium, 2013*; *CARDIoGRAM Consortium, 2010*; *Coronary Artery Disease, C4D*). Genomic risk scores (GRS) have been developed using millions of datapoints (SNPs) and plasma markers. These can contribute to our understanding of critical mechanisms to provide the maximum benefit to the individual, especially before the early stages of pathogenesis. Classical CHD markers defined by the Framingham risk score (FRS), including age, cholesterol, smoking status, blood-pressure, and diabetes status are not predictive in a timely way (*Akilzhanova et al., 2014*). Genetic risk loci have been reported for cell proliferation genes, inflammation and immunity related genes, cholesterol and lipid biogenesis genes, among others. However, only a relatively minor risk could unambiguously be attributed to the wealth of common genetic variants in CHD heritability. Family-based analyses revealed different heritability estimates for distinct sub-phenotypes of CHD (*Fischer et al., 2005*). There is a remarkable consistency in genetic association findings across cohorts (with varying phenotype definitions), underscoring that different manifestations of CHD may have a common genetic architecture (*Lieb &*

*Vasan, 2013*; *Kitsios et al., 2011*). Rare variants with a larger impact and/or more common variants with a smaller biological impact were proposed to cause the observed missing heritability (*Haas et al., 2015*; *Schunkert, Erdmann & Samani, 2010*; *Manolio et al., 2009*).

Ventricular arrhythmias in patients with structural heart disease are responsible for the majority of sudden cardiac deaths (SCD). CAD, previous myocardial infarction is the most common heart disease in which sustained ventricular tachycardia (VT) occurs and reentry is the predominant mechanism. Other cardiac conditions, such as idiopathic DCM, Chagas disease, sarcoidosis, arrhythmogenic cardiomyopathies, and repaired congenital heart disease may also present with VT in follow-up (*Hadid, 2015*).

Recurrent ventricular tachycardia (VT) is an important cause of increased morbidity and mortality in patients with non-ischemic DCM. DCM differs from postinfarction ischemic cardiomyopathy by comprising multiple different etiologies with variable disease progression and prognosis. There is a need for an individualized approach to risk stratification and treatment based on genetic information.

Idiopathic ventricular tachycardia is defined as VT that occurs in patients without structural heart disease, metabolic abnormalities, or long QT syndrome. 10% of all patients referred for evaluation of VT show no obvious structural heart disease. Idiopathic VT is characterized by a structurally normal heart and QRS morphology consistent with site of origin from typical locations of idiopathic ventricular arrhythmias (in particular, the ventricular outflow region). An absence of structural heart disease is usually suggested if an electrocardiogram (ECG) (except in Brugada syndrome and long QT syndrome [LQTS]), echocardiogram, and coronary arteriogram are collectively normal (*Srivathsan et al., 2005*). However, magnetic resonance imaging (MRI) may identify structural abnormalities even if all other test results are normal. Idiopathic VT comprises multiple discrete subtypes that are differentiated by their mechanism, QRS morphology, site of origin, the response to pharmacologic agents, and evidence of catecholamine dependence.

They include right ventricular (RV) monomorphic extrasystoles, RV outflow tract (RVOT) VT, left ventricular (LV) outflow tract (LVOT) VT, idiopathic LV tachycardia (ILVT), and idiopathic propranolol-sensitive (automatic) VT (IPVT). Idiopathic VT from the RVOT and LV are monomorphic and generally not familial. Catecholaminergic polymorphic VT (CPVT), Brugada syndrome, and LQTS are inherited ion channelopathies (*Srivathsan et al., 2005*).

Polymorphic VT may cause syncope and sudden death in Brugada syndrome. Patients with idiopathic VT monomorphic forms have a better prognosis than do patients with polymorphic VT and structural heart disease. Prognosis for patients with VT secondary to ion channelopathies is variable (*Srivathsan et al., 2005*).

Ninety-two patients diagnosed with ventricular tachycardia (VT) with either coronary heart disease (CHD), dilated cardiomyopathy (DCM) or idiopathic ventricular tachycardia (iVT), were enrolled in a study to evaluate the genetic profile and variants in known cardiac risk genes by targeted next generation sequencing (NGS). We hypothesized that shared morphological and phenotypical features among these subgroups might originate from an overlapping molecular basis. In addition, we assumed that the spacious genepool of the population of Kazakhstan that is fueled by more than 100 different ethnic groups deems

this study cohort a challenging but valuable source for interpreting disease-associated genetic variations. Our results provide an important contribution to the understanding of human genetic diversity.

## MATERIALS & METHODS

### Study population

Studies were performed in accordance with the institutional guidelines for human research and the principles of the Declaration of Helsinki. Our research protocol was reviewed and approved by the Ethics Committee of the Center for Life Sciences, National Laboratory Astana, Nazarbayev University and Ethics Committee of the National Research Cardiac Surgery Center (NRCSC), Astana (#20-20/09/17). Written informed consent and permission to publish data was obtained from all research subjects (or their parents for children under 16 years old).

Patients with ventricular tachycardia were enrolled during 2014-2016 at the NRCSC, Astana, Kazakhstan. The study cohort consisted of 92 unrelated patients with ventricular tachycardia (VT) and different background conditions: DCM (DCM VT, $n = 32$), CHD (CHD VT, $n = 23$), and idiopathic VT (iVT, $n = 37$). A clinical diagnosis of patients was verified in all patients by the authors (M.B., O.N., G.R, G.K.) and further experienced cardiologists of the NRCSC, according to international guidelines and criteria (File S1). Patient characteristics and functional parameters are summarized in Table 1 and the statistical testing of clinical parameters is in Table 2. Detailed information on each patient characteristic available in Table S1.

Patients were from Kazakhstan, with Asian and/or Caucasian ancestry. The cohort included sporadic (65/92) and familial (25/92) cases as well as two cases with unknown familial history. The mean age at the time of initial evaluation and diagnosis of the CHD patients was $62.3 \pm 8.7y$ (95% male), $43 \pm 13.3y$ (65.6% male) for the DCM subgroup, and $37.1 \pm 19.2y$ (43.2% male) for the iVT sub-group.

We used sequence data from 60 unrelated Kazakh individuals without known CHD, DCM, and iVT as a comparison group called the Kazakh control group (KCG) representing the general population. The KCG average age was $37.5$ $9\pm 10.9$ years, and the ratio of male:female was 0.63:0.37, respectively. We deposited sequence data from the Kazakh control group in a publicly accessible repository (Submission ID: SUB7590848, BioProject ID: PRJNA646320). Our data are registered with the BioProject database (http://www.ncbi.nlm.nih.gov/bioproject/646320; BioProject ID PRJNA646320).

### Design of the target region for gene enrichment

We designed a custom targeted gene panel using HaloPlex Target Enrichment technology with Agilent Technologies SureDesign software (https://earray.chem.agilent.com/suredesign/). This system allowed us to simultaneously sequence 96 known diagnostic genes for cardiac cardiomyopathies and arrhythmias and additional loci associated with cardiac disorders. HaloPlex technology uses custom molecular inversion probes (SureDesign software, Agilent) for selective circularization-based target enrichment. The diagnostic genes were compiled for 96 genes and target regions that are known causes or candidate

**Table 1** Patient characteristics.

| Characteristics | CHD VT, $n = 23$ | DCM VT, $n = 32$ | iVT, $n = 37$ |
|---|---|---|---|
| Age, years | 62.3 ± 8.8 | 43.0 ± 13.3 | 37.1 ± 19.2 |
| Sex, F/M | 1/22 | 11/21 | 21/16 |
| BMI, kg/m2 | 27.9 ± 5.5 | 27.0 ± 7.2 | 24.9 ± 5.6 |
| NYHA functional class and functional parameters | | | |
| I | 1 (4.3%) | 1 (3.1%) | 25 (67.6%) |
| II | 6 (26.1%) | 1 (3.1%) | 11 (29.7%) |
| III | 16 (69.6%) | 20 (62.5%) | 1 (2.7%) |
| IV | 0 (0%) | 10 (31.2%) | 0 (0%) |
| LVEF, % | 36.6% | 25.5% | 60.9% |
| LA, mm | 42.9 ± 6.2 | 47.3 ± 7.8 | 30.6 ± 6.7 |
| LV EDD, cm | 6.2 ± 1.0 | 6.9 ± 0.8 | 4.6 ± 0.8 |
| LV ESD, cm | 5.1 ± 1.4 | 6,0 ± 0,9 | 3.1 ± 0.9 |
| QRS Interval, ms | 112.4 ± 29.7 | 117.4 ± 27.3 | 89.9 ± 15.4 |
| QT Interval, ms | 401.5 ± 72.0 | 389.0 ± 38.0 | 400.5 ± 44.8 |
| Family history of CM or SCD | | | |
| familial | 6 (26.1%) | 8 (25%) | 11 (29.7%) |
| sporadic | 17 (73.9%) | 24 (75%) | 24 (64.9%) |
| unknown | 0 | 0 | 2 (5.4%) |

Notes.

CHD, coronary heart disease; DCM, dilated cardiomyopathy; VT, ventricular tachycardia; iVT, idiopathic ventricular tachycardia; CM, cardiomyopathy; SCD, sudden cardiac death; LVEF, left ventricle ejection fraction; LA, Left atrial dimension; LV EDD, Left ventricular end-diastolic dimension; LV ESD, Left ventricular end-systolic dimension.

genes for cardiac cardiomyopathies and arrhythmias from PubMed and clinical variant databases (such as HGMD and ClinVar) (File S3). The candidate gene library design covers a total target region of 463.767 kbp (which was used as input for eArray (Agilent Technologies, Santa Clara, California, USA) to design the custom capture-oligonucleotides for in-solution target enrichment with 406.062 analyzable target bases. The analyzable target bases (ATB) included all exonic and proximal intronic (+/-10 bp) sequence information for the 96 cardiac risk genes. ATB are represented by 2,017 target loci.

## Target DNA enrichment and next-generation sequencing

DNA was isolated from fresh-frozen EDTA-blood samples of the patients and processed according to the standard HaloPlex Target Enrichment System Protocol (version D.5, May 2013, Agilent Technologies, Santa Clara, CA, US) using the standard HaloPlex 96 indexing primer cassette. We used the SureSelect Target Enrichment System (Agilent Technologies, Santa Clara, CA, US) for capturing the designed regions. All libraries were quality checked on a 2100 BioAnalyzer (Agilent Technologies, Santa Clara, CA, US) using the High Sensitivity DNA Assay kit, pooled at equimolar amounts and sequenced on a HiSeq2000 platform using 2 ×150 bp paired-end standard sequencing.

**Table 2** Statistical testing (Kruskal–Wallis Test) of clinical parameters.

| | Groups | N | Median | 25-Percentile | 75- Percentile | p-value overall test | Pairwise comparison | $p_{adj}$-value |
|---|---|---|---|---|---|---|---|---|
| LVEF, % | CHD VT | 23 | 35.00 | 26.13 | 45.00 | 0.000 | DCM VT - CHD VT | 0.067 |
| | DCM VT | 32 | 24.50 | 18.25 | 30.00 | | DCM VT - iVT | 0.000 |
| | iVT | 37 | 61.29 | 57.13 | 67.48 | | CHD VT - iVT | 0.000 |
| LA, mm | CHD VT | 23 | 44.00 | 37.00 | 46.00 | 0.000 | DCM VT - CHD VT | 0.537 |
| | DCM VT | 32 | 45.50 | 42.00 | 49.75 | | DCM VT - iVT | 0.000 |
| | iVT | 37 | 31.00 | 26.60 | 35.00 | | CHD VT - iVT | 0.000 |
| LV EDD, cm | CHD VT | 23 | 6.10 | 5.70 | 6.60 | 0.000 | DCM VT - CHD VT | 0.153 |
| | DCM VT | 32 | 6.87 | 6.33 | 7.45 | | DCM VT - iVT | 0.000 |
| | iVT | 37 | 4.70 | 4.22 | 4.97 | | CHD VT - iVT | 0.000 |
| LV ESD, cm | CHD VT | 23 | 4.80 | 4.00 | 5.90 | 0.000 | DCM VT - CHD VT | 0.099 |
| | DCM VT | 32 | 5.92 | 5.63 | 6.50 | | DCM VT - iVT | 0.000 |
| | iVT | 37 | 3.20 | 2.63 | 3.50 | | CHD VT - iVT | 0.000 |
| QRS Interval, ms | CHD VT | 23 | 104.00 | 98.00 | 122.00 | 0.000 | DCM VT - CHD VT | 1.000 |
| | DCM VT | 32 | 113.00 | 98.50 | 122.00 | | DCM VT - iVT | 0.000 |
| | iVT | 37 | 86.00 | 80.00 | 97.00 | | CHD VT - iVT | 0.001 |
| QT Interval, ms | CHD VT | 23 | 400.00 | 374.00 | 450.00 | 0.203 | | |
| | DCM VT | 32 | 394.00 | 363.00 | 403.50 | | | |
| | iVT | 37 | 400.00 | 380.00 | 427.00 | | | |

**Notes.**

CHD, coronary heart disease; DCM, dilated cardiomyopathy; VT, ventricular tachycardia; iVT, idiopathic ventricular tachycardia; CM, cardiomyopathy; SCD, sudden cardiac death; LVEF, left ventricle ejection fraction; LA, Left atrial dimension; LV EDD, Left ventricular end-diastolic dimension; LV ESD, Left ventricular end-systolic dimension.

## Sequencing data processing and variant annotation

Sequence data processing and variant calling was conducted using Agilent NGS data analysis software SureCall version 2.0.7.0 (Agilent Technologies, Santa Clara, CA, USA) with standard settings of the HaloPlex pipeline. Resulting variants were further matched with entries in the Human Gene Mutation Database (HGMD *Stenson et al., 2003*) and annotated with ANNOVAR (*Wang, Li & Hakonarson, 2010*). We included the predictions from the database of human non-synonymous SNVs dbNSFP (*Liu, Jian & Boerwinkle, 2013*) to achieve better scoring. The clinical significance of the sequence variants was interpreted according to the guideline developed by the American College of Medical Genetics and Genomics (ACMG) and the Association for Molecular Pathology (AMP) in 2015 (*Richards et al., 2015*). The ClinVar (*National Center for Biotechnology Information (NCBI), 2019*) and Varsome (*Kopanos et al., 2019*) databases were applied for the variant classification.

The KCG sequencing project identified a total of 2,150 genetic variants in 60 individuals, with a mean coverage of 157-fold at the ATB. The mean coverage of the 92 samples of the cardiology study cohort at the target loci was 707.62-fold and revealed 2,403 distinct genetic variants in the 92 patients. We first eliminated the known common variants with frequencies above 0.5% using a stepwise approach in commonly referenced databases like the ESP6500 or the 1000Genomes Db or SNPDb130. All synonymous variants and variants

observed in the KCG were subtracted from the patient cohort data set, yielding a total of 337 individual non-synonymous variants. The resulting data set, after manual curation, contained 307 individual variants for the overall study population.

## In silico prediction analysis and pathogenicity inference

The pathogenic potential of each variant (HGMD-listed variants and novel or rare variants) was predicted using a combined score from 10 prediction tools: SIFT_score/pred, Polyphen2_HDIVscore/pred, Polyphen2_HVAR_score/pred, LRT_score/pred, MutationTaster_score/pred, MutationAssessor_score/pred, FATHMM_score/pred, RadialSVM_score/pred, LR_score/pred, and MetaSVM_score/pred. Class I (highest pathogenic potential) variants were predicted as being disease-causing by at least 7 of the tools; class II (intermediate pathogenic potential) variants were predicted as being disease-causing by 4–6 of the tools; class III (low pathogenic potential) were predicted as being disease-causing by 1–3 prediction tools; and class IV (benign) was predicted as being disease-causing by none of the tools (0). The PhyloP100 and SiPhy_29 scores are conservation scores and are not designed specifically for finding causal variants for Mendelian diseases, but for finding functionally important sites. Variants that confer increased susceptibility may be scored well. Polyphen2hdiv is commonly used when evaluating rare alleles at loci that are potentially involved in complex phenotypes, dense mapping of regions identified by genome-wide association studies, and analysis of natural selection from sequence data. For further *in silico* analysis, sequence variants were interpreted according to the ACMG/AMP classifications: "pathogenic," "likely pathogenic," "uncertain significance," "likely benign," and "benign". The two sets of criteria were for the classification of pathogenic or likely pathogenic variants and for the classification of benign or likely benign variants (*Richards et al., 2015*). Each pathogenic criterion was weighted as very strong (PVS1), strong (PS1–4), moderate (PM1–6), or supporting (PP1–5), and each benign criterion was weighted as stand-alone (BA1), strong (BS1–4), or supporting (BP1–6). The ClinVar (http://www.ncbi.nlm.nih.gov/clinvar) database was used for its clinical assertions and evidence for the variant classification. Gene symbols recognized by ClinVar were entered and we obtained results with variations affecting the genes. We used the search engine Varsome (https://varsome.com/), which has information from 30 external databases, to look up variant pathogenicity. Pathogenicity of the identified sequence variants is reported using an automatic variant classifier that evaluates the submitted variant according to the ACMG guidelines (*Richards et al., 2015*), classifying it as one of 'pathogenic', 'likely pathogenic', 'likely benign', 'benign' or 'uncertain' significance. We summarized the information about the HGMD-listed variants and the pathogenicity found in the ClinVar, Varsome and final verdict according to the ACMG/AMP guidelines in Table S2.

## Validation of selected mutations

Selected genetic variants (pathogenic mutations, VUS, benign) were reconfirmed using traditional capillary Sanger sequencing (ABI 3730xL Genetic Analyzer; Life Technology, CA, USA) of the PCR product for all suspected samples. Primers were used for preliminarily

determined mutations (Fig. S5, Table S7). Mutation fragments were amplified using DNA Taq polymerase (Takara, Japan). PCR conditions consisted of 1 cycle of 96 °C for 5 min, 30 cycles of 96 °C for 2 min, 55 °C or 57 °C for 30 s, 72 °C for 1 min; 1 extension cycle of 72 °C for 5 min and holding at 4 °C.

## Statistics

A standard quality control (QC) protocol was applied to eliminate implausible data points and outliers. We used the Shapiro Wilk test to determine normality of the distribution ($p > 0.05$ normally distributed data assumed) and Q–Q plots. We performed a non-parametric Kruskal-Wallis test for variables that did not meet the assumption of normality. The chi-square test was used to compare categorical variables and the Fisher exact test was used for $2 \times 2$ contingency tables, if the expected count was less than 5. Data are presented as total number (%), and for skewed distributions, as median and Interquartile range (25-percentile and 75-percentile). All statistical tests were performed using SPSS version 23.0 (SPSS Inc., Chicago, IL). A two tailed $p$-value of less than 0.05 was considered as statistically significant.

# RESULTS

Ninety-two unrelated patients with ventricular tachycardia (VT) and either coronary heart disease (CHD), dilated cardiomyopathy (DCM) or idiopathic ventricular tachycardia (iVT) were prospectively enrolled for genetic analyses to evaluate their genetic profile and variation in known cardiac risk genes (*Bekbossynova et al., 2018*; *Akilzhanova et al., 2019*).

The clinical characteristics of the patients in the three clinical subgroups are summarized in Table 1 and listed in detail in Table S1. The predominant NYHA functional classes were II-III for CHD, III-IV for DCM, and I-II for the iVT subgroup. LVEF, LA, LV ESD, and LV EDD significantly differed between the iVT and the two other subgroups (Table 2). The proportion of familial cases was nearly the same in all three subgroups, ranging from 25–29.7%.

## Genetic variants

Targeted enrichment and sequencing, variant calling and stepwise filtering of the annotated variants identified a total of 307 unique variants in 74 genes totaling up in 456 variants for the overall study group. The frequency and pathogenicity of variants associated with the different arrhythmogenic syndromes within the different subgroups studied is shown in Table S2. Variants included: one in/del variant, four splice-site variants, and 451 single-nucleotide variants (SNV) within the coding exonic regions. Seven (0.15%) of the SNVs were unique stop-gain variants, three of which were residing in the TTN gene. 168 HGMD mutations (61 unique) were observed in 37 genes. 33% of the HGMD mutations were predicted to have an increased pathogenic potential (class I and II variants); 49% were classified class III variants and 18% of the observed HGMD variants were classified as benign by all prediction algorithms (Fig. 1). In contrast to the HGMD variants, a higher proportion (40%) of the novel and rare variants (≤0.5% in ESP6500 or 1000G db) was predicted to be pathogenicity classes I and II (Fig. 1).

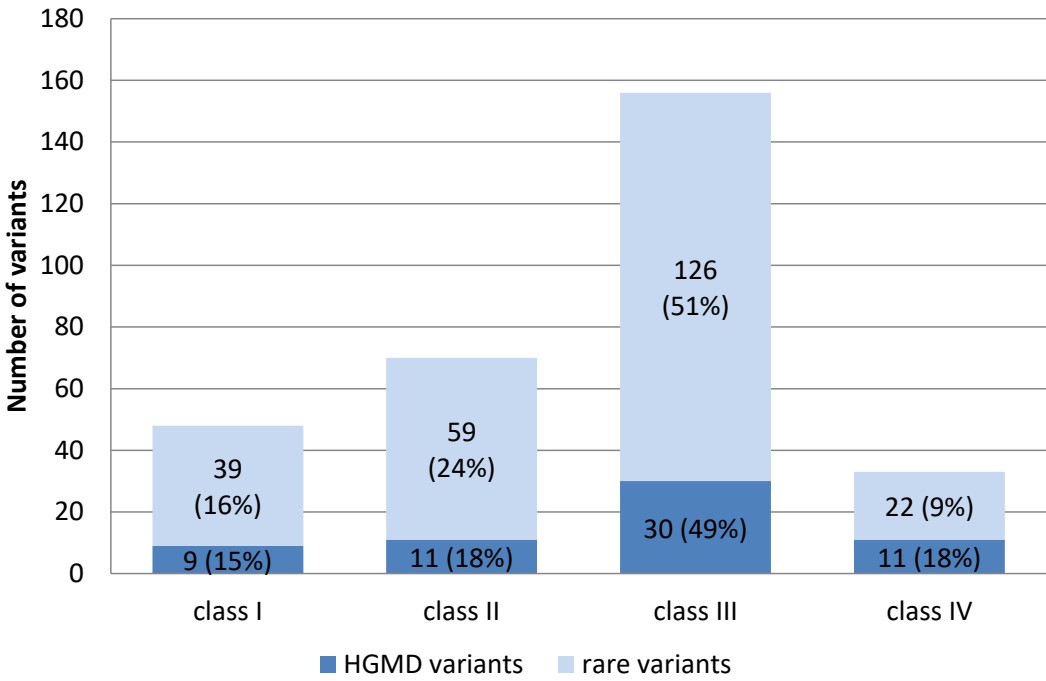

**Figure 1  Distribution of HGMD listed variants and rare variants with respect to their pathogenic potential.** Mean average variant frequency in public databases (ESP6500, 1000G2012apr_all, and EXAC_ALL) is 0.000279 for class I variants (high pathogenic potential), 0.00267 for class II (intermediate pathogenic potential), 0.00372 for class III (low pathogenic potential), and 0.0192 for class IV (benign) variants.

Variants with the highest pathogenicity score (class I variants) made up approximately 8.7% of the CHD VT subgroup, and 6.3% and 18.9% in the DCM VT and the iVT groups, respectively (Fig. 2). The prevalence of class II variants was moderately lower (30.4%) in the CHD VT group for the DCM VT and the iVT groups (31.4% and 43.2%, respectively).

Statistical testing of and/or rare variants between the different clinical subgroups (testing was done including and omitting titin variants) indicated no difference in the average number of HGMD variants or rare variants alone between the three subgroups (Table S3, Fig. S1A, Fig. S1B).

Classification of 307 variants according to ACMG guidelines showed that 9 (2.9%) variants were classified as pathogenic, 9 (2.9%) were likely pathogenic, 98 (31.9%) had uncertain significance, 73 (23.8%) were likely benign and 118 (38.4%) were benign (Fig. 3, Table S2). ACMG pathogenic and likely pathogenic variants were observed in classes III and IV, and contrary benign and likely benign variants were observed in class I and class II variants. Most variants were variants of uncertain significance.

## Measure of molecular burden

We used the cumulative potential pathogenic variance to determine whether there was a difference in the genetic variation per patient, as a measure of the molecular burden, within

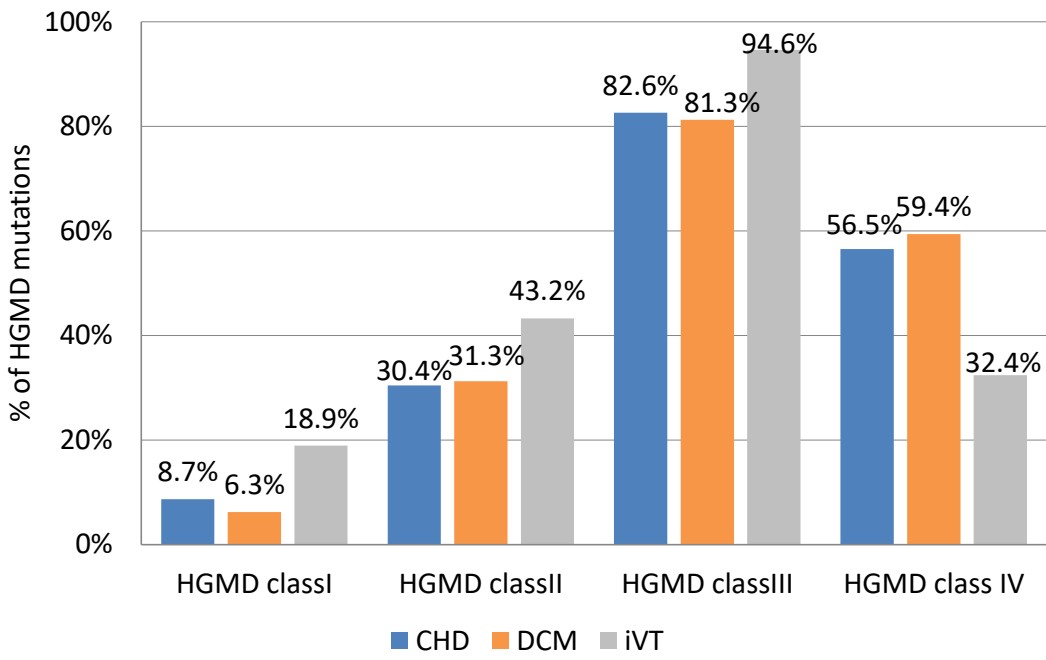

**Figure 2** **Frequency of HGMD mutations per clinical subgroup.** CHD, coronary heart disease; DCM, dilated cardiomyopathy; iVT, idiopathic ventricular tachycardia; HGMD, Human Gene Mutation database. Class I (highest pathogenic potential) variants were predicted disease causing by at least 7 of the tools, class II (intermediate pathogenic potential) variants were predicted disease causing by 4–6 of the tools, class III (low pathogenic potential) were predicted disease causing by 1–3 prediction tools, and class IV (benign) was predicted disease causing by none of the tools (0)

the three different subgroups (Table 3). 10 patients with CHD VT (43.5%) carried at least one class I variant, averaging 1.3 class I variants per positive individual. The same variant frequency was observed for the DCM VT and the iVT subgroups (31.3%; 10 individuals carrying 13 class I variants and 20 individuals carrying 26 class I variants, respectively). 69.6% (CHD VT), 75% (DCM VT), and 81% (iVT) of the patients carried about two class I or II (intermediate pathogenic potential) variants. The inclusion of class III variants (low pathogenic potential) increased the average variant frequency to 4.05 (CHD VT; 95.7% of patients), 4.23 (DCM VT; 96.9%), and 4.44 (iVT; 97.3%) per patient (Table 3, Fig. S2).

None of the CHD VT patients and only one DCM VT and iVT patient carried zero HGMD or other rare class I-IV variants (Table S4). More than 75% (CHD VT: 86.9%, DCM VT: 78.13%, iVT: 81%) of the patients carried at least one HGMD mutation, irrespective of the disease group.

## Distribution of the functional effects of detected mutations

We evaluated whether there might be a differential distribution of variants between the disease groups in relation to their functional context. We grouped the genes into seven categories using information from GeneCards® and The Human Gene Database https://www.genecards.org/ (cell membrane, cytoskeleton, sarcomere, metabolism, intercalated disc, ion flux, and nucleus) based on their molecular function and/or

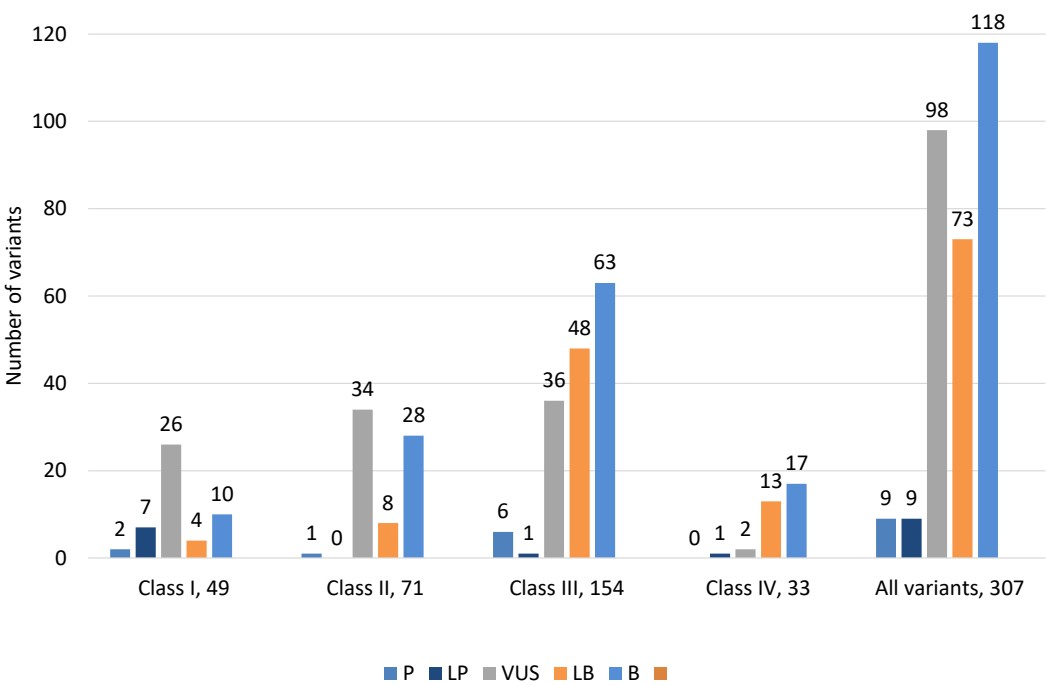

**Figure 3** **Distribution of variants according to ACMG guidelines among all classes of 307 genetic variants.** Class I (highest pathogenic potential) variants were predicted disease causing by at least 7 of the tools, class II (intermediate pathogenic potential) variants were predicted disease causing by 4–6 of the tools, class III (low pathogenic potential) were predicted disease causing by 1–3 prediction tools, and class IV (benign) was predicted disease causing by none of the tools (0). The ACMG/AMP classification: P, pathogenic; LP, likely pathogenic; VUS, variant of uncertain significance; LB, likely benign and B, benign.

**Table 3** **Frequency of patients positive for pathogenetic variants in the clinical subgroups.**

| | patients carrying ≥1 class I variant | % positive | cumulative number of variants | % of all variants | variants per positive patient[a] |
|---|---|---|---|---|---|
| CHD VT ($n = 23$) | 10 | 43.5% | 13 | 11.8% | 1.3 |
| DCM VT ($n = 32$) | 10 | 31.3% | 13 | 8.2% | 1.3 |
| iVT ($n = 37$) | 20 | 54.1% | 26 | 13.9% | 1.3 |
| | ≥ 1class I/II variant | | | | |
| CHD VT ($n = 23$) | 16 | 69.6% | 31 | 28.2% | 1.94 |
| DCM VT ($n = 32$) | 24 | 75.0% | 49 | 30.8% | 2.04 |
| iVT ($n = 37$) | 30 | 81.1% | 70 | 37.4% | 2.33 |
| | ≥ 1 class I/II/III variant | | | | |
| CHD VT ($n = 23$) | 22 | 95.7% | 89 | 80.9% | 4.05 |
| DCM VT ($n = 32$) | 31 | 96.9% | 131 | 82.4% | 4.23 |
| iVT ($n = 37$) | 36 | 97.3% | 160 | 85.6% | 4.44 |

**Notes.**

[a]Variants per positive patient was calculated by dividing the (cumulating) number of variants by the number of positive patients. Class I (highest pathogenic potential) variants were predicted disease causing by at least 7 of the tools, class II (intermediate pathogenic potential) variants were predicted disease causing by 4–6 of the tools, class III (low pathogenic potential) were predicted disease causing by 1–3 prediction tools, and class IV (benign) was predicted disease causing by none of the tools (0).

**Table 4   Distribution of class I–IV variants according to their molecular function/association.** Class I (highest pathogenic potential) variants were predicted disease causing by at least seven of the tools, class II (intermediate pathogenic potential) variants were predicted disease causing by 4–6 of the tools, class III (low pathogenic potential) were predicted disease causing by 1–3 prediction tools, and class IV (benign) was predicted disease causing by none of the tools (0).

|  | Class I | Class II | Class III | Class IV |
|---|---|---|---|---|
| Cell membrane | 0 (0%) | 7 (41.2%) | 8 (47.1%) | 2 (11.8%) |
| Cytoskeleton | 5 (9.8%) | 12 (23.5%) | 22 (43.1%) | 12 (23.5%) |
| Sarcomere | 14 (11.7%) | 34 (18.3%) | 62 (51.7%) | 10 (8.3%) |
| Metabolism | 4 (23.5%) | 2 (11.8%) | 10 (58.8%) | 1 (5.9%) |
| Intercalated disc | 6 (20.0%) | 5 (16.7%) | 15 (50.0%) | 4 (13.3%) |
| Ion flux | 12 (30.8%) | 7 (17.9%) | 18 (46.2%) | 2 (5.1%) |
| Nucleus | 7 (21.2%) | 3 (9.1%) | 21 (63.6%) | 2 (6.1%) |

subcellular association. Distribution of class I–IV variants according to their molecular function/association are shown in Table 4. There was a moderate, but statistically insignificant, underrepresentation of variants in the cell membrane (4.3%) genes and intercalated disc (4.3%) genes in CHD VT patients compared to the other groups (DCM VT: 9.4 and 12.5; iVT: 10.8% and 21.6%, respectively) when we included only class I and II variants in the analysis (Fig. 4 and Fig. S3 and Table S5). Variants in the metabolism-associated genes were moderately overrepresented in the CHD VT subgroup (13%) compared to the DCM VT subgroup (3.1%) alone.

Based on their relative frequencies HGMD mutations in *LAMA2* (34.3%), *MYBPC3* (31.2%), *MYH6* (18.7%), *KCNQ1* (15.6%), *GAA* (15.6%) and *DSG2* (12.5%) were predominant in the DCM VT subgroup (Fig. 5). The mutation and variant distribution of the CHD VT subgroup strongly overlapped with the patterns for the other subgroups (Table S2, Table S6, and Fig. S4). *PRKAG2* mutations p.G100S ($n = 3$, 13%, CM136115) and novel p.H222Q variant were observed in four CHD VT patients (Table S2). Statistical testing suggested a trend towards an increased frequency of *PRKAG2* variants in the CHD subgroup (CHD VT: 13.04% vs. DCM VT: 3.1% and iVT: 2.7%; *p*-value 0.053) (Fig. S4). There was a prevalence of iVT mutations in genes encoding ion flux.

There were 9 pathogenic variants of ACMG, including W746C in *GAA* in patients with CHD, R218Q *KCNJ2* and R5338X *TTN* in patients with iVT, and F244L *MYH7*, Q353X *LMNA*, L17465X *TTN*, W21011X *TTN*, c.2334+1G >A *DSG2*, c.477+1G >A *KCNQ1* in patients with DCM (Table S2). Sanger sequencing confirmed some of the observed genetic variants. (Fig. S5).

## Genetic variants in the healthy Kazakh group

From a total of 2,150 variants, observed in the practically healthy individuals, KCG ($n = 60$), 475 were common polymorphisms and thus subtracted from further analysis. The remaining 1,675 variants included 68 exonic variants (37 synonymous, three frameshift deletions, three in/del non-frameshift deletions, and 25 non-synonymous single nucleotide variants). Variants with a MAF (minor allele frequency) of ≥0.5% in the ESP6500 or the 1000G yielded 58 exonic variants were also excluded. 3 (5.2%) were predicted to be class I,
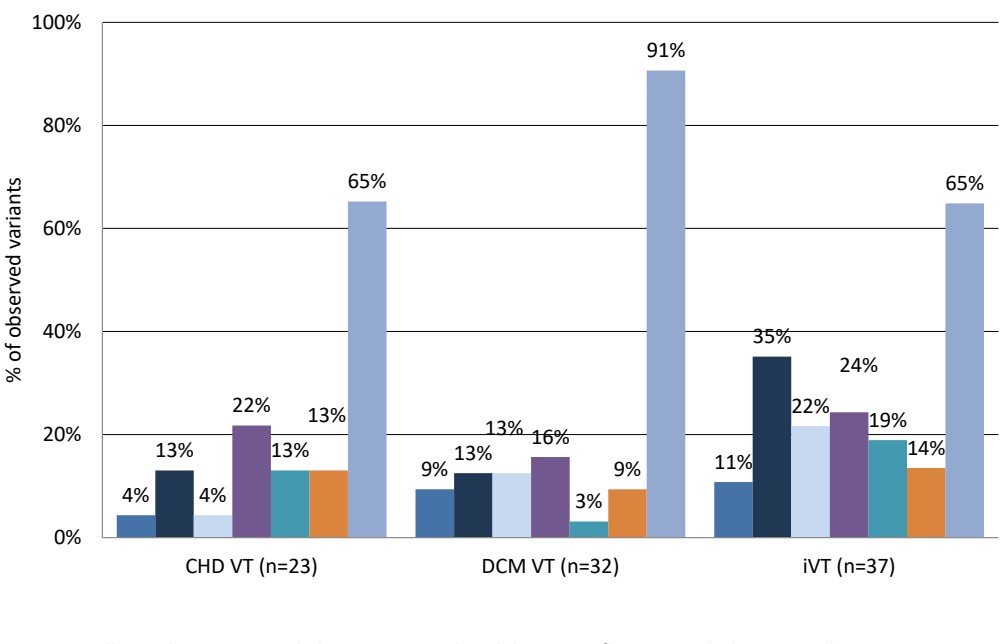

**Figure 4 Distribution of variants (class I+class II) according to their molecular function/association.** The genes were grouped into seven categories using information from GeneCardsR and The Human Gene Database https://www.genecards.org/ (cell membrane, cytoskeleton, sarcomere, metabolism, intercalated disc, ion flux, and nucleus) based on their molecular function and/or subcellular association. Combined mutations of class I and class II were included. Class I (highest pathogenic potential) variants were predicted disease causing by at least 7 of the tools, class II (intermediate pathogenic potential) variants were predicted disease causing by 4–6 of the tools. CHD, coronary heart disease; DCM, dilated cardiomyopathy; iVT, idiopathic ventricular tachycardia; VT, ventricular tachycardia.

11 (19%) were class II, 36 (62%) were class III, and 8 (13.8%) were predicted to be class IV. Thus, the average frequency of a class I variant in the KCG ($n = 60$) was 5%. We analyzed the presence and frequency of 307 genetic variants found in patients in the KCG group. 58 genetic variants were observed in KCG and 5 of these genetic variants was a mutant minor allele (Table S2).

## DISCUSSION

We evaluated the contribution of molecular genetic variants in genes associated with cardiac disorders in Kazakhstani population. We identified a significant proportion of possible pathogenic variants using molecular genetic screening with a targeted next-generation sequencing (NGS) panel (*Bekbossynova et al., 2018*; *Akilzhanova et al., 2019*). We obtained data on the distribution of genetic variants, the number of mutations, and the mutational burden of patients with ventricular tachycardia of various etiology.

NGS technologies have emerged as an efficient alternative to Sanger-sequencing, providing the analytical characteristics for the comprehensive exploration of genetic mechanisms (*Frese, Katus & Meder, 2013*; *Roberts et al., 2013*). Furthermore, it is believed that NGS will be increasingly important in the studies of monogenic and complex

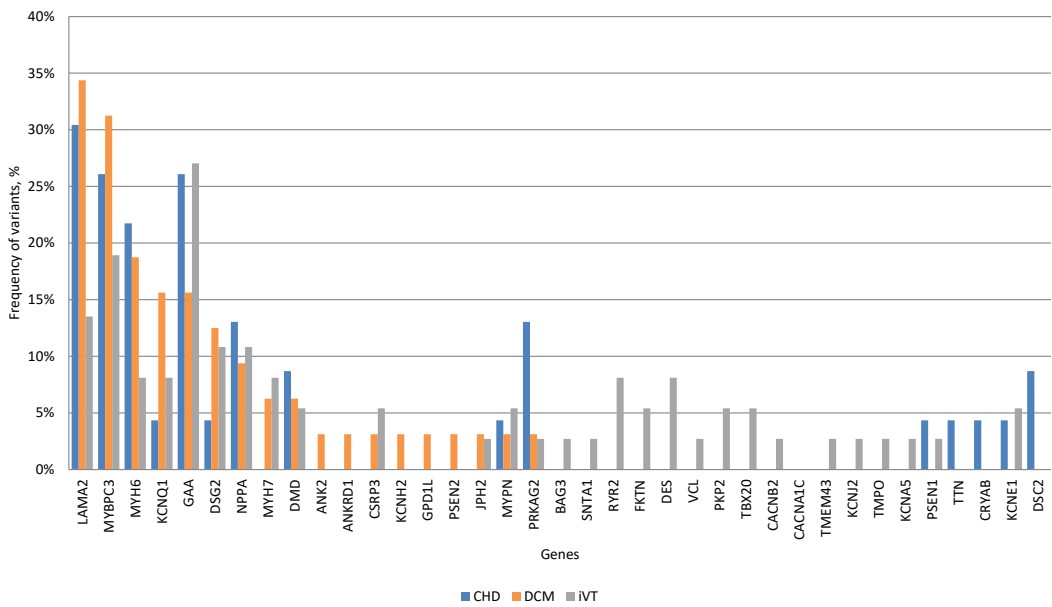

**Figure 5** **Frequency of HGMD plus variants within each clinical subgroup.** CHD, coronary heart disease; DCM, dilated cardiomyopathy; iVT, idiopathic ventricular tachycardia; HGMD; Human Gene Mutation database.

diseases, such as common cardiovascular diseases (CAD, cardiomyopathies and others) in which one or more variants in a single gene, or multiple variants in different genes, are involved (*Haas et al., 2015*; *Meder et al., 2011*; *Frese, Katus & Meder, 2013*; *Roberts et al., 2013*). The ability of NGS to generate high-throughput qualitative and quantitative sequence information has enabled investigations that were previously technically infeasible or cost prohibitive (*Schrijver et al., 2012*).

There are some disadvantages to the use of NGS, including the incomplete representation and coverage of exons, which poses the risk of limiting sensitivity and the inability to detect clinically significant mutations. Targeted enrichment of certain genes followed by the use of NGS for high-throughput genetic testing of genes for heart disorders is now becoming feasible and technically proven. This has been evidenced by the almost complete coverage and high accuracy of the approach, offering greater sequencing depth with reduced costs and data burden (*Meder et al., 2011*; *Schrijver et al., 2012*).

We sequenced three groups of patients with CHD VT, DCM VT and VT of unknown etiology (idiopathic) in our study to identify genetic variants that were associated with the three cardiovascular phenotypes and to evaluate the level of genetic variation in cardiac risk genes in these distinct subgroups. We designed and optimized a custom target-enrichment assay of 96 genes associated with cardiac disorders using Haloplex technology (Agilent Technologies, Santa Clara, USA) (*Bekbossynova et al., 2018*; *Akilzhanova et al., 2019*).

Targeted enrichment and sequencing and stepwise filtering of the annotated variants identified a total of 307 unique variants in 74 genes totaling up in 456 variants for the overall study group. The filtering step is crucial for bioinformatics analysis to reduce the

number of probable and potentially pathogenic variants such as the exclusion of common variants present in the Single Nucleotide Polymorphism database (dbSNP). Filtering is based on assumptions about the attributes of the disease-causing variant(s), including the effect of the variant on the protein, the presumed absence of the variant in the dbSNP database, or the frequency cutoffs based on minor allele frequency from the 1000 Genomes Project (*Schrijver et al., 2012*).

The DCM and other common congenital heart disorders affect approximately 1–4 people per 10,000 population. We selected a MAF cut-off of 0.5% for rare variants in order to balance the rate of false positives (non-pathogenic variants) at the cost of losing variants with moderate/intermediate pathogenicity (true positives) that provoke increased susceptibility for complex diseases like CHD. With the introduction of multigene panels, exome sequencing, and whole-genome sequencing, the numbers of variants identified per person has increased progressively, and disease mutation databases contain potentially benign variants that were previously classified as disease causing. The interpretation of genetic variants is complex.

We used a combined score of 10 prediction tools to determine the pathogenic potential of each variant (HGMD listed variants and novel or rare variants). Class I (highest pathogenic potential) variants were predicted as being disease causing by at least 7 of the tools; class II (intermediate pathogenic potential) variants were predicted as being disease causing by 4–6 of the tools; class III (low pathogenic potential) were predicted as being disease causing by 1–3 prediction tools; and class IV (benign) was predicted as being disease causing by none of the tools (0).

As expected, the MAF was the lowest for the variants with the most severe pathogenicity (class I) (0.000279), followed by intermediate pathogenicity variants (class II) with an MAF of 0.00267 being nearly a 10-fold higher than that for class I. The average MAF for the variants with the lowest pathogenicity (class III variants with low pathogenic potential) was relatively common at 0.00372.

The pathogenicity of sequence variants was classified using an automatic variant classifier according to the ACMG guidelines. The classifications were: 'pathogenic', 'likely pathogenic', 'likely benign', 'benign', or 'uncertain significance'. Information about HGMD listed variants and the classification of pathogenicity found in the ClinVar, Varsome and final verdict according to the ACMG/AMP were summarized in Table S2.

We classified 307 variants according to ACMG guidelines, which showed that 9 (2.9%) variants were pathogenic, 9 (2.9%) were likely pathogenic, 98 (31.9%) had uncertain significance, 73 (23.8%) were likely benign and 118 (38.4%) were benign (Fig. 3). ACMG pathogenic and likely pathogenic variants were observed in classes III and IV, and contrary benign and likely benign variants were observed in class I and class II variants. Most variants had uncertain significance.

Our results show that complex methods are required to make a final interpretation of sequenced variants.

Patients of all subgroups were clinically diagnosed according to common international classification criteria. Despite the many common genetic risk variants identified for CHD in GWAS, they only account for a small percentage of the expected heritability (*Roberts et al.,*

*2013*; *Qian & Bodmer, 2012*). The predisposition to CHD is estimated to be approximately 50% genetic, although the 36 variants identified by CARDIoGRAM and the follow-up CARDIoGRAMplusC4D project only accounted for about 10% of the heritability. Rare risk variants with minor allele frequencies ≤1–5%, complex gene-gene interactions (epistasis), and undiscovered common variants are thought to cause this discrepancy. We sequenced 23 individuals with CHD VT and observed a spectrum of genetic variation that quantitatively (frequency of genetic variants) and qualitatively (molecular function) strongly overlapped with DCM VT and iVT. 43.5% (10/23) of CHD VT patients carried a class I variant, whereas only 5% (3/60) of the control cohort (KCG), 31.3% (13/32) of the DCM VT and 54.1% (20/37) of iVT patients carried a class I variant. If class II variants were added, 69.5% (16/23) of CHD VT, 75% (24/32) of DCM VT and 81.1% (30/37) of iVT patients carried on average two variants of high to intermediate pathogenicity. High- to intermediate pathogenicity variants in *LAMA2, MYBPC3, MYH6, KCNQ1, GAA,* and *DSG2* predominated in CHD VT patients at similar frequencies as those observed for DCM VT and iVT patients. This similarity points to a common molecular disease-association. Independent of multiple tested disease-associated mutations (HGMD) and rare or newly identified variations with increased pathogenic potential, there was no statistically significant difference in the frequency of genetic variation between the three subgroups. Our results confirmed that DCM and iVT patients frequently carry multiple mutations or variants with high pathogenic potential, which has been shown in previous research. The high frequency of rare genetic variants with increased pathogenic potential (class I and II) in CHD patients was unexpected. The commonly referenced concepts of cardiomyopathies as monogenic disorders has been challenged (*Haas et al., 2015*; *Qian & Bodmer, 2012*; *Lopes et al., 2013*), indicating complex interactions of genes and the significance of rare variants. Distinct clinical phenotypes, including LQT, Brugada syndrome and HCM, revealed that multiple mutations, and rare potential pathogenic and functional variants in affected individuals could synergistically or additively alter penetrance, age-of-onset, or disease progression (*Meder et al., 2011*; *Kapa et al., 2009*; *Stattin et al., 2012*; *Allegue et al., 2015*).

*TTN, GAA, LAMA2* and *MYBPC3* harbored the most variants in the three subgroups which confirm the high impact of these genes in complex pathogenesis of cardiomyopathies and VT demonstrated in previous studies (*Herman et al., 2012*; *Golbus et al., 2012*; *Bit-Avragim et al., 2001*; *Carboni et al., 2011*).

Classification of the variants according to their cellular function showed that sarcomere function, ion-flux, nuclear function, and metabolism were predominantly affected by variants with the highest pathogenic potential (class I variants) in CHD VT patients. A similar pattern was observed for class I variants in DCM VT and iVT patients. In addition, iVT patients carried variations potentially affecting the cytoskeleton and intercalated disc. Pooling class I-III variants yields metabolism-associated variants in >60% of CHD VT patients, which is second behind variants in sarcomere genes. On a gene basis, *PRKAG2* mutations were overrepresented in the CHD sub-group versus DCM and iVT, (*p*-values = 0.053 and 0.054, respectively). The mean age of the four heterozygous mutation carriers was 67.7 years (+/- 5.1y). *PRKAG2* encodes the γ2 regulatory subunit of the AMP-activated protein kinase AMPK and mutation-associated defects account for a

cardiac syndrome triad consisting of familial ventricular preexcitation (*Gollob et al., 2001*), conduction system disease, and cardiac hypertrophy mimicking (HCM) (*Gollob et al., 2002*), with a significant proportion of those progressing to DCM. HCM-associated *PRKAG2* mutations are generally not associated with myocyte and myofibrillar disarray, which are the pathognomonic features of HCM, but with pronounced vacuole formation within myocytes due to excessive glycogen accumulation (*Aggarwal et al., 2015*; *Arad et al., 2002*). This may be explained by the central regulatory function of AMPK during acute low-energy states in which ATP-consuming pathways are shut off, like glycogen, cholesterol and fatty acid synthesis and the ATP-producing pathways are enhanced, such as fatty acid oxidation and glucose uptake. The potential functional role of AMPK in atherosclerosis has recently been shown by the protective effect of melatonin on the cardiovascular system, since flow shear stress-induced apoptosis in bone marrow mesenchymal stem cells could be reversed via the activation of AMPK (*Yang et al., 2016*).

## CONCLUSIONS

We showed that in patients with VT, secondary to coronary artery disease, DCM, and idiopathic etiology, multiple rare mutations and clinically significant sequence variants in classic cardiac risk genes associated with cardiac channelopathies and cardiomyopathies were found in a similar pattern and at a comparable frequency. CHD VT patients were found to carry rare genetic variants with an increased pathogenic potential at a comparable frequency as DCM VT and iVT patients. These variants were found in genes related to sarcomere function, nuclear function, ion flux, and metabolism. Our study size was limited but this pilot study suggests that monogenic diseases can serve as an insightful model for complex disorders. Patients with coronary heart disease, dilated cardiomyopathy, and idiopathic ventricular tachycardia share overlapping patterns of pathogenic variation in cardiac risk genes. A greater in-depth statistical analysis like sub-grouping of participants according to other features like ethnicity, severity, or anamnesis (familial vs. sporadic, etc.) was not possible at this stage due to the group size limitation. Additional studies including more patients with and without ventricular tachycardia will be needed to generate a deeper insight into genotype-phenotype correlation of CHD and cardiomyopathies and between idiopathic ventricular tachycardia types.

## ACKNOWLEDGEMENTS

We thank A. Groselj-Strele at the Core Facility Computational Bioanalytics (Med. Univ. of Graz) for statistical analysis.

### Funding

This work was supported by the Ministry of Education and Science of the Republic of Kazakhstan on projects (O.0703, N 0115RK01931; AP05134683; AP05136106, AP05134722,

AP05134737). The funders had no role in study design, data collection and analysis, decision to publish, or preparation of the manuscript.

### Grant Disclosures

The following grant information was disclosed by the authors:
Ministry of Education and Science of the Republic of Kazakhstan on projects: O.0703, N 0115RK01931, AP05134683, AP05136106, AP05134722, AP05134737.

### Competing Interests

The authors declare there are no competing interests.

### Author Contributions

- Christian Guelly conceived and designed the experiments, performed the experiments, analyzed the data, prepared figures and/or tables, authored or reviewed drafts of the paper, designed gene panel, and approved the final draft.
- Zhannur Abilova, Katrin Panzitt, Ainur Akhmetova, Saule Rakhimova and Ulan Kozhamkulov performed the experiments, prepared figures and/or tables, and approved the final draft.
- Omirbek Nuralinov and Galina Kaussova performed the experiments, analyzed the data, prepared figures and/or tables, patients recruitment and diagnosis, and approved the final draft.
- Ulykbek Kairov analyzed the data, prepared figures and/or tables, authored or reviewed drafts of the paper, and approved the final draft.
- Askhat Molkenov and Ainur Ashenova analyzed the data, prepared figures and/or tables, and approved the final draft.
- Slave Trajanoski analyzed the data, prepared figures and/or tables, designed gene panel, and approved the final draft.
- Gulzhaina Abildinova (Rashbayeva) performed the experiments, analyzed the data, prepared figures and/or tables, and approved the final draft.
- performed the experiments, analyzed the data, prepared figures and/or tables, patients recruitment and diagnosis, and approved the final draft.
- Christian Windpassinger conceived and designed the experiments, analyzed the data, prepared figures and/or tables, gene panel design, and approved the final draft.
- Joseph H. Lee analyzed the data, authored or reviewed drafts of the paper, and approved the final draft.
- Zhaxybay Zhumadilov conceived and designed the experiments, analyzed the data, authored or reviewed drafts of the paper, and approved the final draft.
- Makhabbat Bekbossynova conceived and designed the experiments, performed the experiments, analyzed the data, authored or reviewed drafts of the paper, patients recruitment and diagnosis, and approved the final draft.
- Ainur Akilzhanova conceived and designed the experiments, performed the experiments, analyzed the data, prepared figures and/or tables, authored or reviewed drafts of the paper, designed gene panel, and approved the final draft.

## Human Ethics

The following information was supplied relating to ethical approvals (i.e., approving body and any reference numbers):

The research protocol was reviewed and approved by the Ethics Committee of the Center for Life Sciences, National Laboratory Astana, Nazarbayev University and Ethics Committee of the National Research Cardiac Surgery Center (NRCSC), Astana (approval # 20- 22/09/17); and written informed consent and permission to publish data was obtained from all research subjects or their parents in the case of children under 16 years old.

## Data Availability

Data is available at NCBI Sequence Read Archive (SRA): PRJNA382749. Sequence data from the Kazakh Control cohort is also available at SRA: PRJNA646320. Data analyzed in this study are available in the Supplemental Files.

## Supplemental Information

Supplemental information for this article can be found online at http://dx.doi.org/10.7717/peerj.10711#supplemental-information.

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
