# Peer review of "Patients with coronary heart disease, dilated cardiomyopathy and idiopathic ventricular tachycardia share overlapping patterns of pathogenic variation in cardiac risk genes"

_PeerJ, doi:10.7717/peerj.10711_

## Round 0.1 · original submission · Major Revisions

First, I apologize for the unusual delay in getting back to you with the reviews in order for you to undertake revisions. You may respond to the different points of the reviewers and defend an alternate point of view with supported arguments. However, I agree with Reviewer 1 that it would be optimal to validate a subgroup of the variants identified by the NGS panel using Sanger sequencing, as false-positives are always a concern, and the panel means that the findings are hypothesis-driven. As Reviewer 2 remarks, given the philosophy of Open Science that PeerJ espouses https://peerj.com/about/policies-and-procedures/#data-materials-sharing), please deposit the sequence data from the Kazakh Control cohort to a publicly accessible repository for other scientists to also use for analysis.

Reviewer 1 ·

Basic reporting

The paper presents genetic informations with low impact for clinical cardiologists due to excessive technical details with not clear clinical utility. It is fundamental that the authors furnish more clinical information and modify some parts of genetic analysis (opinion of genetic section of my hospital). See below

Experimental design

see below

Validity of the findings

see below

Additional comments

Comments for genetic analysis:

1. Authors investigated the cohort of patients by using a NGS panel of 96 genes. The libraries were quality checked by using a BioAnalyzer. However, the diagnostic sensitivity and specificity of this 96 genes panel was not tested. Therefore, for at least a subgroup of variants identified by the NGS panel, their presence should be confirmed by Sanger sequencing.

2 - The potential pathogenicity of the identified variants was investigated by using two distinct classifications. The first consists of a combined score as a result of 10 predictor tools (lines 221-234). By using this, authors classify variants in four classes (from Class I "highest pathogenetic potential" to Class IV "benign"). The second classification is quite common and widely used, the ACMG/AMP classification (lines 234-251). These two classifications gave different results. In fact, as authors state "ACMG pathogenic and likely pathogenic variants were observed in class III, IV and contrary benign and likely benign variants were observed in class I and class II variants". Looking at Figure 3, it appears that 38 variants classified as benign by the ACMG/AMP classification, are present in Class I (highest pathogenetic potential) and in Class II (intermediate pathogenetic potential). It would be better to use the ACMG/AMP classification only.

3 - How authors evaluate the frequency of the distinct classes of mutations in the healthy individuals (KCG group) is confusing. In fact, from the 2150 variants found in KCG group, authors subtract those in common wth the patient group (475). Why they do this? By doing this it is very likeli that authors reduced in the control group the frequency of Class I and II mutations. Therefore, the average frequency of Class I variant in the KCG group is probably higher than the 3.3% claimed by authors.

Clinical observations:

1. A risk stratification assembling VT due to structural heart disease and idiopathic VT is not so clear. The pathogenesis of the different VT forms is totally different and the authors should differentiate more clearly the different diagnostic role of genetic analysis in the various etiologies of VT.

2 Moreover, no information are reported for the different types of idio VT: they include totally different forms (RVOT VT, fascicular VT, aortic VT, etc). More details are fundamental for this aspect

·

Basic reporting

The introduction needs to be expanded to include VT, which is mentioned in the title, the abstract, the discussion and is even one of the keywords, yet is not addressed in the introduction while much space is given to DCM and CHD.

Unfortunately, the WHO reference (number 1) does not seem to be valid anymore and hence cannot be easily accessed. There is a big typo in line 102 with a WHO link that could relate to this. It would be important to find a reference that is readily accessible.

Experimental design

No comment.

Validity of the findings

No comment.

Additional comments

The research question is relevant in the context of human genetics due to an underrepresentation of the population studied in this work (Central asian) in many genetic studies (GWAS and otherwise). While most databases have expanded to incorporate cohorts from a defined geographical origin, coverage of other genetically distinct and/or highly heterogeneous populations has lagged behind. In the context of the genetics of heart disease, the data presented here suggest a common genetic origin to different presentations of VT.

While the percentage of predicted pathological variants is relatively low (less than 6%), the panel was based on known cardiac risk genes hence sampling a relatively small yet clinically relevant group of genes. In addition, most patients (~70%) carried at least one variant from the HGMD with predicted high or intermediate pathogenic potential.

The raw data is available in the NCBI SRA (ID: PRJNA382749), and the analysed data is available in the supplementary material (e.g. Table S2). There is one piece of raw data that it’s not clear if it is available, the sequence data from the 60 Kazakh unrelated individuals (the KCG). The text mentions it as "unpublished sequencing data...".

In the section Distribution of the functional effects of detected mutations, line 327, it says "Based on their molecular function and/or subcellular associations, we assigned the genes into seven groups...". Where was the information obtained from? If a database, which? Was Gene Ontology used? This information would be useful for readers.

The figures highlight the presented data in a clear, understandable manner.

There are a number of typos throughout the text that should be corrected before publication, examples in lines 161 ("...characteristic..."), 241 ("...databases..."), 244 ("...search engine namely Varsome"), 328 ("...we assigned the genes into seven groups...").

Also, formatting errors in lines 91, 92 and 102.

---

## Round 0.2 · Minor Revisions

I am prepared to accept the article if you can have it re-read by a native English speaker in order to ready the copy. The authors may choose between an independent English editing service or a colleague/peer, but should provide a certificate or confirmation that they have performed this English editing service.

·

Basic reporting

I appreciate the effort the authors have put into addressing the comments from the reviewers (particularly considering the ongoing health situation), and I think the manuscript now closely aligns with PeerJ's publishing standards.

Experimental design

Nothing to add to the previous comments.

Validity of the findings

The authors have not modified the findings in a substantial way to warrant a modification of my positive opinion on the validity of the findings.

Additional comments

There is still a significant number of typos throughout the manuscript:

Line 50 "...were included into study with aim to..."
Line 71 "...subgroups studied which is confirm high impact of this genes in complex pathogenesis of ..."
Line 96 "...deaths are occur in..."
Etc

The manuscript should be proofread in detail before publication.

---

## Round 0.3 · accepted · Accept

Thank you for the final corrections and your contribution to the literature in human genetics. I have suggested a few copy-editing annotations.